# Contextual Transformer for Offline Reinforcement Learning

## Abstract

Recently, the pretrain-tuning paradigm in large-scale sequence models has made significant progress in Natural Language Processing and Computer Vision. However, such a paradigm is still hindered by intractable challenges in Reinforcement Learning (RL), including the lack of self-supervised large-scale pretraining methods based on offline data and efficient fine-tuning/prompt-tuning over unseen downstream tasks. In this work, we explore how prompts can help sequence-modeling based offline Reinforcement Learning (offline-RL) algorithms. Firstly, we propose prompt tuning for offline RL, where a context vector sequence is concatenated with the input to guide the conditional generation. As such, we can pretrain a model on the offline dataset with supervised loss and learn a prompt to guide the policy to play the desired actions. Secondly, we extend the framework to the Meta-RL setting and propose Contextual Meta Transformer (CMT), which leverages the context among different tasks as the prompt to improve the performance on unseen tasks. We conduct extensive experiments across three different offline-RL settings: offline single-agent RL on the D4RL dataset, offline Meta-RL on the MuJoCo benchmark, and offline MARL on the SMAC benchmark. The results validate the strong performance, and generality of our methods.

## 1 Introduction

Reinforcement learning algorithms based on sequence modeling (Chen et al., 2021; Janner et al., 2021; Reed et al., 2022) shine in sequential decision-making tasks and form a new promising paradigm. Compared with classic RL methods, such as policy-based methods and value-based methods (Sutton & Barto, 2018), optimization of the policies from the sequence prospective has advantages in long-term credit assignment, partial observation, etc. Meanwhile, significant generalization of large pretrained sequence model in natural language processing (Kenton & Toutanova, 2019; Brown et al., 2020) and computer vision (Liu et al., 2021b; Zhai et al., 2021) not only conserves vast computation in downstream tasks but also alleviates the large data quantity requirements. Inspired by them, we want to ask the question: *whether pretrain technique has a similar power in RL?* Since limited and expensive interactions impede the deployment of RL in various valuable applications (Levine et al., 2020), pretraining a large model to improve the robustness of real-world gap by a zero-shot generalization and improve data efficiency by few-shot learning offers great significance. (Team et al., 2021; Meng et al., 2021) pretrains a single model with diverse and abundant training tasks in the decision-making domain to generalize in downstream tasks, which proves the feasibility that pretraining enables RL agents to harness knowledge for generalization.

Earlier works on sequence modeling RL provide a new perspective on offline RL. However, extending these methods to the pretrain domain is still confronted with several challenges. One major challenge for generalization (Li et al., 2020b) is how to encode task-relevant information, thereby enhancing transferring knowledge among tasks. Since discovering the relationship among diverse tasks from data and making decisions conditioned on distinct tasks plays a significant role in generalization, it is not a trivial modification of existing methods. Another problem is efficient self-supervised learning in offline RL. Specifically, the decision transformer (Chen et al., 2021) leverages the data to learn a return conditioned policy, which ignores the knowledge about world dynamics. In addition, trajectory transformer (Janner et al., 2021) conducts planning based on a world model, but the high computational intensity and decision latency might be a bottleneck for a large-scale model and hard to fine-tune to other tasks. Therefore, introducing key components to transfer the knowledge in a

pretrained model and incorporating the advantages in a conditioned policy and a world model is necessary.

In this work, we propose a novel offline RL algorithm, named **C**ontextual **M**eta **T**ransformer (CMT), aiming to conquer multiple tasks and generalization at one shot in an offline setting from the perspective of sequence modeling. CMT provides a pretrain and prompt-tuning paradigm to solve offline RL problems in the offline setting. Firstly, a model is pretrained on the offline dataset through a self-supervised learning method, which converts the offline trajectories into some policy prompts and utilizes these policy prompts to reconstruct the offline trajectories in the autoregressive style. Then a better policy prompt is learned based on planning in the learned world model to attain the advanced policy to generate trajectories with high rewards. In contrast to previous work, CMT learns a prompt to construct policy to guide desired actions, rather than being designed by humans or explicitly planned by the world model. In the offline meta-learning setting, CMT extends the framework by simply concatenating a task prompt with the input sequence. With a simple modification, CMT is capable of executing a proper policy for a specific task and sharing knowledge among tasks.

Our contributions are three-fold: First, we propose a novel offline RL algorithm based on prompt tuning, in which the offline trajectory is encoded as a prompt, and the appropriate prompt is found to lead a policy for execution to achieve high reward in the online environment. Second, CMT is the first algorithm to solve offline meta-RL from a sequence modeling perspective. The context trajectory, which represents the structure of the task, is used by CMT as a prompt to guide the policy in a specific unknown task. Furthermore, empirical results on D4RL datasets and meta Mujuco tasks show that CMT has outstanding performance and strong generalization.

## 2 RELATED WORK

**Offline Reinforcement Learning.** Offline RL is gaining popularity as a data-driven RL method that can effectively utilize large offline datasets. However, the data distribution shift and hyper-parameter tuning in offline settings seriously affect the performance of the agent (Levine, 2021). So far, several schemes have been proposed to address them. Through action-space constraint, BCQ (Fujimoto et al., 2019), AWR (Peng et al., 2019), BRAC (Wu et al., 2019), and ICQ (Yang et al., 2021) reduce extrapolation error caused by policy iteration. Noticing the problem of overestimation of values, CQL (Kumar et al., 2020) keeps reasonable estimates by looking for pessimistic expectations. UWAC (Wu et al., 2021) handles out-of-distribution (OOD) data by weighting the Q value during training by estimating the uncertainty of $(s, a)$. MOPO (Yu et al., 2020) and MOReL (Kidambi et al., 2020) solve the offline RL problem from the model-based perspective while ensuring rational control by adding penalty items to uncertain areas. Decision Transformer (DT) (Chen et al., 2021) and Trajectory Transformer (TT) (Janner et al., 2021) reconstruct the RL problem into a sequential decision problem, extending the Large-Language-Model-like (LLM-like) structure to the RL area, which inspires many follow-up works on them. However, the relevant works on offline RL are still insufficient due to the lack of self-supervised large-scale pretraining methods and efficient prompt-tuning over unseen tasks, and CMT proposes a pretrain-and-tune paradigm to deal with them.

**Pretrain and Sequence Modeling.** Recently, much attention has been attracted to pretraining big models on large-scale unsupervised datasets and applying them to downstream tasks through fine-tuning. In language process tasks, transformer-based models such as BERT (Kenton & Toutanova, 2019), GPT-3 (Brown et al., 2020) overcome the limitation that RNN cannot be trained in parallel and improve the ability to use long sequence information, achieving SOTA results on NLP tasks such as translation, question answering systems. Even the CV field has been inspired to reconstruct their issues as sequence modeling problems, and high-performance models like the swin transformer (Liu et al., 2021b) and scaling ViT (Zhai et al., 2021) have been proposed. Since the trajectories in offline RL datasets have Markov properties, they can be modeled through transformer-like structures. Decision transformer (Chen et al., 2021) and trajectory transformer (Janner et al., 2021) propose condition policy on return to go (RTG) and behavior cloning policy improved by beam search to solve RL problems in offline settings respectively. Inspired by these works, we bring prompt tuning from NLP into the RL domain, then propose a potential path to pretrain a large-scale RL model and efficiently transfer the knowledge to downstream tasks.

**Offline meta-RL and Task Generalization.** Offline meta-RL shines recently since it allows algorithms to adapt to new tasks quickly without interacting with the environment. Targeting it, an optimization-based method with advantage weighting loss called MACAW (Mitchell et al., 2021) is proposed, which learns the initialization of both the value function and the policy. FOCAL (Li et al., 2020b) combines the deterministic context encoder with behavior regularization and achieves inspiring results based on an off-policy Meta-RL method called PEARL (Rakelly et al., 2019). Then it is improved by combining the intra-task attention mechanism and the inter-task contrastive learning objective, which is named FOCAL++ (Li et al., 2021), to deal with sparse reward and distribution shift. BOReL (Dorfman et al., 2020) aims to learn Bayesian optimal policies from discrete data for the mentioned problems, whereas SMAC (Pong et al., 2021) learns meta-policies from reward-labeled data and then fine-tunes on new tasks. From the model-based perspective, MerPO (Lin et al., 2022) proposes a meta-model for efficient task structure inference and a meta-policy for safe exploration of OOD data. It is worth mentioning that recent work on general model construction, such as SayCan (Ahn et al., 2022), and Gato (Reed et al., 2022), has achieved exciting results, demonstrating the huge potential of LLM-like architectures. Just like them, CMT is also a general LLM-like model that can solve offline meta-RL problems effectively.

## 3 PRELIMINARY

**Meta Reinforcement Learning.** The major purpose of meta-RL is to leverage multi-task experience to enable fast adaptation to new unseen tasks. A task $\mathcal{T}_i$ is defined as a Markov Decsion Process (MDP) $\mathcal{T}_i = (\mathcal{S}, \mathcal{A}, \mathcal{R}, \mathcal{P}, \lambda)$, where $\mathcal{S}$ is the state space, $\mathcal{A}$ is the action space, $\mathcal{R}$ is reward function, and $\mathcal{P}$ is transition function. In deep RL, the policy $\pi_\theta(a_t|s_t)$, which specifies the probability that the agent takes action $a_t$ in state $s_t$ at time $t$, is described by a neural network with parameters $\theta$. The goal in a MDP is to learn a optimal policy $\pi^* = \arg\max_\pi \mathbb{E}_{s_0, a_0, s_1, a_1, \dots}[\sum_{t=0}^{\infty} \lambda^t r(s_t, a_t)]$ which can maximize the expected discounted return, where $\lambda$ is a discounted factor. In meta-RL, tasks are drawn from a task distribution $\mathcal{T}_i \sim p(\mathcal{T})$, the state space $\mathcal{S}$ and the action space $\mathcal{A}$ are common across tasks, and reward function $\mathcal{R}_i$ and transition function $\mathcal{P}_i$ are task specific. During meta-training, the meta policies are trained based on some training tasks sampled from task distribution to achieve fast adaptation to new unseen tasks in meta tests.

**Offline Reinforcement learning.** In offline RL setting, the trajectory dataset $D$ is collected from unknown behavior policy $\mu$, which might be an expert policy, sub-optimal policy, random policy, or a mixture policy (e.g. corresponding the replay buffer of an RL agent). A offline trajectory $\tau$ consists of states, actions, and scalar rewards: $\tau = \{\mathbf{s}_t, \mathbf{a}_t, r_t\}_{t=0}^{T-1}$. A trajectory fragment $\tau_{[t_1:t_2]}$ denotes transitions from time-step $t_1$ to time-step $t_2$. This paper aims to learn an optimal policy $\pi^*$ from the fixed dataset $D$ without interaction with the environment.

**Prompt and Prompt-Tuning.** Conditional generation tasks are common in NLP, where the input is a context $x$ and the output $y$ is a sequence of tokens. Autoregressive model LM (Brown et al., 2020) is a powerful tool to solve this kind of tasks, which concatenates the context and the output as a whole sequence $u = [x, y]$ and models the probability for the next token $u_i$ based on the previous tokens $u_{<i}$:

$$h_i = \text{LM}(u_i, h_{<i}),$$
$$p(u_i|u_{<i}) = \text{softmax}(Wh_i), \tag{1}$$

where $u_i$ denotes $i$-th token in the sequence $u$, $h_i \in \mathbb{R}^d$ denotes the activation in transformer at time step $i$, and $W$ is the learning parameter matrix. To leverage the knowledge in the pretrained large-scale model, prompts are designed to improve the few-shot performance in the downstream task. A prefix-style prompt $z$, also a sequence of tokens, are concatenated with input $u = [z, x, y]$ to guide the model to generate the desired output. Besides hand-designed prompts $z$, prompt-tuning(Li & Liang, 2021) is proposed to learn a continuous prompt that can be learnt from data.

## 4 METHOD

In this section, we introduce CMT, an RL framework for offline RL and offline meta-RL. We describe CMT with policy prompts for offline RL in Section 4.1, and CMT extended with task prompts and policy prompts for meta-RL in Section 4.2.

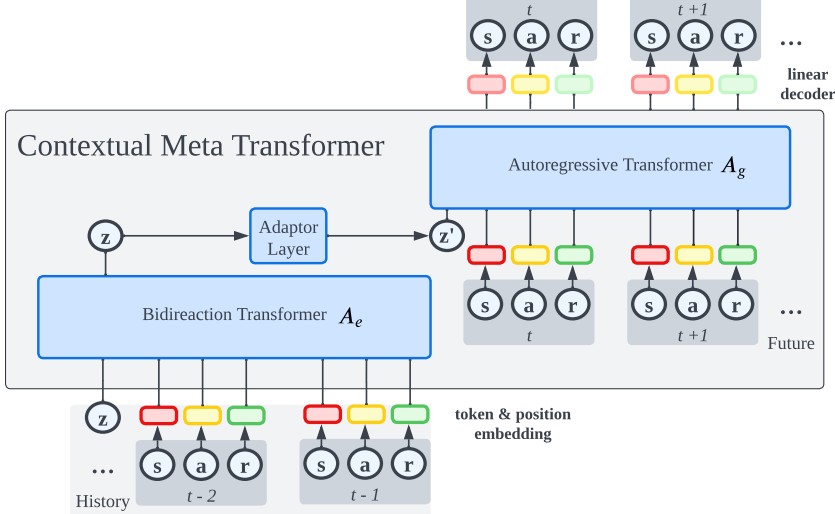

Figure 1: The framework for CMT in the offline learning Setting. (a) In the representation stage, CMT pretrains an auto-encoder model in the offline dataset, which predicts the future action, reward, and state with the policy prompt from the history trajectory. The adaptor layer is a identity function during this stage, which mean $z \equiv z'$. (b) In the improvement stage, we freeze the pretrianed model, and tune the prompt to reach a better performance.

## 4.1 OFFLINE SEQUENCE LEARNING

The main assumption of our method is that offline trajectories can be viewed as samples from unknown policies, and the optimal policy can be represented as a mixture of these basic policies. Our method contains two stages of training, the representation stage and the improvement stage. CMT learns a model to convert an offline trajectory into a policy prompt with some characteristics to represent these deterministic policies. In the second stage, a policy prompt is learnt to mix up basic policies by planning in the world model to attain an advanced policy.

**Representation Stage.** Fig.1 shows the whole architecture, which constitutes an auto-encoder $A$ in trajectory-level. CMT consists of two components: a trajectory encoder $A_e$ with parameter $\theta$ and an autoregressive generator $A_g$ with parameter $\phi$. Trajectory encoder $A_e$ is a bi-direction transformer (Kenton & Toutanova, 2019), which gets a history trajectory and gives the policy prompt $z_\tau$ for the trajectory $z_\tau = A_e(\tau; \theta)$. Autoregressive generator $A_g$ is a GPT-style (Brown et al., 2020) conditional generator, which predicts the policy prompt $z_\tau$ and the next token in the future based on the previous history trajectory: $\tau_{t+1} = A_g(.|z_\tau, \tau_{<t}; \phi)$.

In this stage, CMT introduces two loss terms to update $\theta$ and $\phi$. The major loss $\mathcal{L}_1$ is supervised loss, which is used to reconstruct the whole trajectory, and an auxiliary loss $\mathcal{L}_2$ help improve policy in the next stage. The loss is linear weighted as $\mathcal{L} = \mathcal{L}_1 + \gamma \mathcal{L}_2$, in which $\gamma$ is the contrastive loss coefficient. For the supervised loss $\mathcal{L}_1$, since $A_g$ predicts the future action, reward and state one by one, it employs as an union of a policy $\pi(a|s) = A_g(z_\tau, s_t, \tau_{<t})$, a dynamic model $P(s'|s, a) = A_g(z_\tau, \tau_{<t})$ and a reward function $R(s, a) = A_g(z_\tau, a_t, s_t, \tau_{<t})$. The prediction and the ground truth form a supervised loss in Eq.(2):

$$\mathcal{L}_1(\tau; \phi, \theta) = \sum_{t=0}^{T-1} (\mathcal{D}(s_t, P(\tau_{<t}; \phi, \theta)) + \mathcal{D}(a_t, \pi(s_t, \tau_{<t}; \phi, \theta)) + \mathcal{D}(r_t, R(a_t, s_t, \tau_{<t}; \phi, \theta)), \quad (2)$$

in which distance matrices $\mathcal{D}$ adopts MSE loss for deterministic output and cross-entropy loss for stochastic prediction. Since the entire architecture is differentiable, the supervised loss can be used to update $A_e$ and $A_g$.

An auxiliary loss $\mathcal{L}_2$ constrains the distance between prompts coming from similar trajectories by self-supervised learning. Inspired by (Liu et al., 2021a), an effective and stable policy improvement based on imitation learning often satisfies two properties: (a) Keeping new behavior close to pre-

vious ones. (b) Getting higher rewards than the previous ones. As we desire to improve the policy by prompt tuning, it is natural to facilitate the similarity of prompts from similar trajectories. For this purpose, we introduce an InfoNCE contrastive loss (Van den Oord et al., 2018) to constrain the prompt in a self-supervised method to meet the aforementioned requirements. The auxiliary contrastive loss is given as Eq.(3):

$$\mathcal{L}_2(\tau_q, \{\tau_i\}_{i=1}^K; \theta) = -\log \frac{\exp(A_e(\tau_q; \theta) \cdot A_e(\tau_+; \theta)/\alpha)}{\sum_{i=1}^k \exp(A_e(\tau_q; \theta) \cdot A_e(\tau_i; \theta)/\alpha)} = -\log \frac{\exp(z_q \cdot z_+/\alpha)}{\sum_{i=1}^k \exp(z_q \cdot z_i/\alpha)}, \tag{3}$$

in which $\alpha$ is temperature coefficient. For the anchor policy prompt $z_q$ encoded from trajectory $\tau_q$, a batch of $K$ policy prompts $\{z_i\}_{i=1}^K$ encoded from a set of trajectories $\{\tau_i\}_{i=1}^K$ sampled from the offline dataset. $\{z_i\}$ consists of $K-1$ negative samples $z_-$ and one positive sample $z_+$. The definition of the positive and negative samples influence the property of the policy prompt. To ensure similar behavior trajectories be encoded into close prompts, the auxiliary loss defines the pair of policy prompts samples from the same trajectory and different trajectories as the positive and negative sample pair.

**Improvement Stage.** Since the behaviour policy can be sub-optimal in the offline dataset, we consider prompt tuning to boost the agent performance, with the purpose to transfer the knowledge in the pretrained model. As shown in Fig.(1), the key idea is simple: we can freeze the pretrained model, and learn prompts that can guide the pretrained model to generate a trajectory with high reward. Specifically, improvement stage consists of relabeling the offline dataset and prompt tunning by adaptor layer.

Relabeling the offline dataset is to replace the raw ordinary action with better action to provide new supervised target for prompt tunning. Concretely, we sample a mini-batch of data, and then adopts the beam search method proposed by trajectory transformer Janner et al. (2021) as a planning algorithm to find the better action, in which the autoregressive generator $A_g$ works as a world model.

To improve the performance by prompt tunning, we should tune the policy prompt $z_\pi$ for a better policy prompt $z'_\pi$ to guide generator $A_g$ to generate a trajectory with a higher reward. For this purpose, we freeze the pretrained model parameters, denoted as $\bar{\theta}$ and $\bar{\phi}$ and only tune the parameter $\xi$ for the adaptor layer $L$ on the relabeled dataset. The adaptor layer $L$ is trained by the following Eq.(4),

$$\mathcal{L}_3(\tau; \xi) = \sum_0^{T-1} \mathcal{D}(\hat{a}_t, \pi(s_t, \tau_{<t}; \bar{\phi}, \bar{\theta}, \xi)) + \beta(z - L(z; \xi))^2 \tag{4}$$

in which $\hat{a}_t$ is the relabeled action, and the second term constrains behavior changes to alleviate distribution shift in the offline setting, like (Fujimoto & Gu, 2021) and $\beta$ is a weight coefficient for behavior constraint. This method can be regarded as using prompt tunning to remember planning results in the world model, which significantly reduce the computation cost and decision delay in evaluation. However, it should be noticed that we use planning method as the improvement method, but any other improvement algorithm of which loss function based on the output of generator $A_g$ can be easily plugged in.

## 4.2 Contextual Sequence Meta Learning

Extended from the section 4.1 which introduces policy prompts to solve offline RL problem, we simply incorporates a task prompts in CMT to achieve generalization ability in downstream unseen task in offline meta RL setting. The task encoder $F(t|\tau)$ is used to encode transitions into a task hidden variable $t$ and learn a contextual policy $\pi(a|s, t)$ in classical context meta-RL methods. Therefore, CMT is feasible to extend to the meta-learning domain by simply plugging in task prompts. Fig.(3) shows the minor modifications supporting CMT have impressive generality.

**Meta Training.** To contain the task information, CMT simply concatenates a context trajectory, which is a trajectory fragment coming from the same task, with the input. During offline meta-training, the context trajectory is randomly sampled from the offline dataset. To avoid information fusion, we separate the context and history trajectories with a special token ([SEP]), whose parameters can be learned. Then we adopt a contrastive learning method similar to Eq.(3) to learn a stable and consistent task prompt, similar to (Fu et al., 2021) in online meta-RL. The major difference

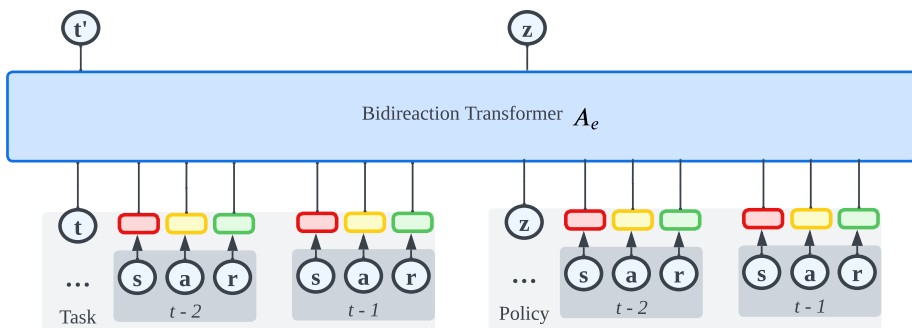

Figure 2: The framework of CMT in Offline meta reinforcement Learning Setting. Based on the basic framework in Figure.1, CMT introduces a context trajectory as task prompt in trajectory encoder $A_e$ to guide $A_g$.

between contrastive loss in task prompts and policy prompts is that task prompts form positive and negative sample pairs in task-level, while policy prompts form positive and negative sample pairs at trajectory-level.

**Meta Test.** After training on diverse tasks, meta test stage requires agent rapidly adapting in the unseen task. In context meta RL, agent is permitted to collect few context trajectories to understand the task. The context trajectory in the meta test could come from an offline dataset in an unknown task or a trajectory that has interacted with the online world. The second setting is more challenging (Dorfman et al., 2021) due to the exploration problem. To verify the strong capacity of CMT, we evaluate CMT in the second setting. Furthermore, CMT discards recursive component, so it is suitable for zero-shot setting, which means CMT collects the context during online evaluation, rather than in advance. To the best of our knowledge, there is no existing method to solve this one-shot setting in offline meta RL. As a result, we construct a context buffer to store the history of interactions, and the context trajectories are randomly chosen from the context buffer.

## 5 EXPERIMENT

In this section, we evaluate the performance of CMT in terms of offline RL tasks in D4RL benchmarks (Fu et al., 2020), offline meta-RL tasks in meta Mujoco benchmarks (Todorov et al., 2012). Additional offline multi-agent experiments are conducted on StarCraft II Micromanagement (Pong et al., 2021). Simply replacing a sequence of transition by a sequence of agent, CMT can be easily extended to solve multi-agent offline-RL tasks and is evaluated in a popular MARL benchmark (SMAC). The results on SMAC are reported in Appendix A.3. Apart from the performance in various settings, we design experiment for ablation study to demonstrate the validity of the components contained in CMT. Our experiments are conducted on a server with Nvidia Tesla A100 GPU and AMD EPYC 7742 CPU.

### 5.1 OFFLINE LEARNING TASKS

We evaluate CMT on the continuous control tasks from D4RL benchmarks. The experiments on four standard Mujoco locomotion environments (HalfCheetah, Hopper, Walker, and Ant) are conducted with three kinds of dataset quality (Medium, Medium-Replay, and Medium-Expert). The differences between them are as follows: **Medium** contains 1 million timesteps generated by a "medium" policy interacting with the environment, with an intelligence level of around 1/3 that of experts. **Medium-Replay** contains the replay buffer generated during the medium policy training process, and about 25k-400k timesteps are included in the tested environments. **Medium-Expert** consists of 1 million timesteps generated by the medium policy concatenated with another 1 million timesteps generated by the expert policy.

Five baselines are considered, including behaviour cloning (BC) (Torabi et al., 2018), behavior regularized ActorCritic (BRAC) (Wu et al., 2019), conservative Q-learning (CQL) (Kumar et al.,

2020), implicit Q-learning (IQL) (Kostrikov et al., 2021), and decision transformer (DT) (Chen et al., 2021). BC realizes intelligence by learning from expert datasets, which is actually a supervised learning process that learns the states to predict actions. Because of severe extrapolation errors caused by the policy evaluation, traditional offline RL algorithms perform poorly. And the methods such as BCQ, BRAC, and IQL, avoid extrapolation errors by constraining the behavior space. While CQL solves it by finding a conservative Q function that keeps the policy function's expected value less than the true value. Starting from another perspective, DT transforms the RL problems into sequence modeling problems and attempts to find the optimal actions. The detail about hyper-parameter lists is in Appendix. A.2.

The results for D4RL datasets are shown in Table. 1, CMT performs excellently on the Medium and Medium-expert datasets, but not so well on the Medium-replay dataset, indicating that CMT prefers to learn from data generated by stable policies. Compared with DT, which is also a transformer-based structure, CMT outperforms it in most of the tasks. Moreover, although IQL is the SOTA algorithm currently, the performance of CMT on the Medium and Medium-expert datasets meets or exceeds it, demonstrating that our method has huge potential.

Table 1: Results for D4RL datasets. Here we report the mean for three seeds, and the reward is normalized so that 100 represents an expert policy and 0 represents a worst policy in D4RL. PT abbreviation stands for prompt tunning. In addition, our method name and the best performances are bold font.

| Dataset | Environment | **CMT** with PT | **CMT** w/o PT | DT | BRAC-v | CQL | IQL | BC |
|---|---|---|---|---|---|---|---|---|
| Medium-Expert | halfcheetah | **92.9** | 59.8 | 88.0 | 41.9 | 91.6 | 86.7 | 65.6 |
| Medium-Expert | hopper | **106.5** | 102.0 | 103.3 | 0.8 | 105.4 | 91.5 | 55.4 |
| Medium-Expert | walker | 97.6 | 83.5 | 108.4 | 81.6 | 108.8 | **109.6** | 11.2 |
| Medium-Expert | ant | 101.3 | 67.1 | 89.3 | - | 115.8 | **125.6** | 71.2 |
| Medium | halfcheetah | 43.6 | 40.1 | 42.1 | 46.3 | 44.0 | **47.4** | 41.6 |
| Medium | hopper | **68.9** | 62.8 | 62.0 | 31.1 | 58.5 | 66.3 | 48.6 |
| Medium | walker | 75.0 | 69.6 | 71.6 | 81.1 | 72.5 | **78.3** | 47.8 |
| Medium | ant | 71.8 | 61.3 | 64.6 | - | 90.5 | **102.3** | 63.7 |
| Medium-replay | halfcheetah | 38.7 | 16.5 | 36.3 | **47.7** | 45.5 | 44.2 | 2.2 |
| Medium-replay | hopper | 84.9 | 58.4 | 67.8 | 0.6 | **95.0** | 94.7 | 30.8 |
| Medium-replay | walker | 49.5 | 37.3 | 47.8 | 0.9 | 26.7 | **73.9** | 5.9 |
| Medium-replay | ant | 40.6 | 42.1 | 61.7 | - | **93.9** | 88.8 | 30.1 |

## 5.2 Offline Meta Learning Tasks

We explore four tasks to evaluate CMT on zero-shot generalization: Half-Cheetah-Vel, Ant-Fwd-Back, and Ant-Fwd-Back. The same data collection method is used as described in the literature (Li et al., 2020b). The following baselines are taken into account: **Batch PEARL** (Rakelly et al., 2019): A modified version of PEARL which can be used for offline RL tasks. **CBCQ** (Fujimoto et al., 2019): An advanced version of the BCQ that has been adapted to offline RL tasks by incorporating latent variables into state information. **MBML** (Li et al., 2020a): A multi-task offline RL method with metric learning. **FOCAL** (Li et al., 2020b): A model-free offline Meta-RL method with state-of-the-art performance based on the deterministic context encoder. These baselines are trained on a set of offline RL tasks and are tested on the set of unseen offline RL tasks.

The results for meta Mujoco environments are shown in Fig. 3. Once again, we should emphasize the results of CMT is zero-shot setting, while all the other baselines requires context from offline dataset or online interactions in advance. As we can see, CMT can outperform most baselines, including CBCQ, batch PEARL, and MBML. Besides, FOCAL is the SOTA algorithm currently, while CMT can outperform it in different tasks except Walker-2D-Params, showing that our algorithm also has great potential in the area of offline meta-RL.

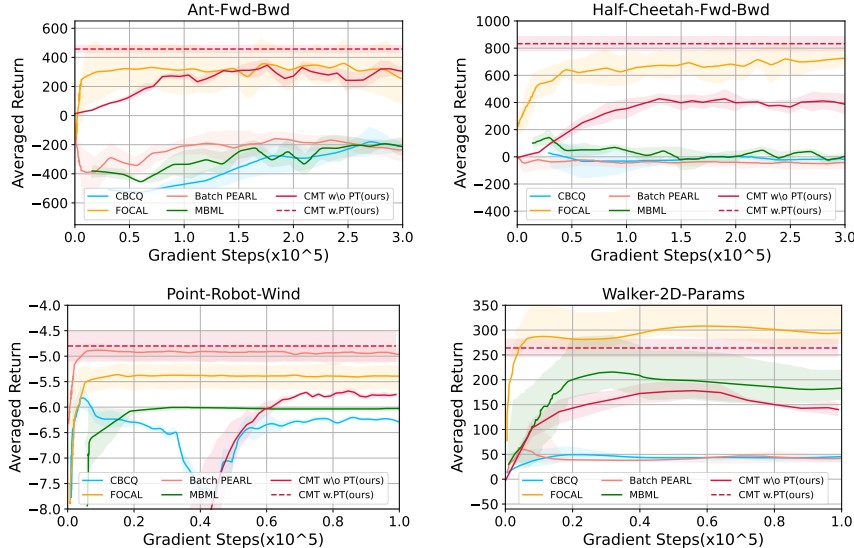

Figure 3: Results for Meta Mujoco Environment. In all benchmark tasks, CMT obviously learns that a policy can face adaptation into a new task, and provides evidence that sequence modeling method is promising. Noticed that CMT have two training stage, it is difficult to align the x-axis. Therefore, we report the training curve of CMT in representation stage and the final evaluation results of CMT after prompt-tunning as a dotted line with standard deviation.

## 5.3 ABLATION STUDY

In this section, we formulate experiments to investigate the following research questions: **Q1:** How important prompt-tunning is for performance? **Q2:** Does contrastive loss benefit the prompt-tuning? **Q3:** Does the behavioral constraint affect the results? **Q4:** In offline meta RL setting, does quality of task prompts affect the performance in downstream tasks? Without loss of generality, we completed the following ablation experiments in the Ant-Fwd-Bwd environment, and the results are shown in Figure 5.

**(Q1) Prompt-tuning.** Prompt-tuning is utilized in the second stage to enhance the model's performance based on the pre-trained model. With the help of it, the average return will increase by 65.8%, which shows that prompt-tuning is effective in improving model effects. Furthermore, the results of CMT with prompt-tuning and without prompt tuning in Table 1 and Figure 3 strongly demonstrates the benefit in performance from improvement stage.

**(Q2) Contrastive Loss.** Contrastive loss plays a key role in clustering similar trajectories, which ensures that the model can find the correct prompts to guide better trajectories. To investigate its influence, we utilized two extreme coefficients, the minimal value of $10^{-6}$ and the maximal value of 1. As shown in Figure 5, when the coefficient gets to its minimal value, the average return drops by 50.9%. A possible explanation is that the model lacks the clustering process, preventing it from generating effective prompts to distinguish different types of trajectories. Finally, the model will be heavily affected by data distribution shifts during the tuning process. When the coefficient gets maximal, the average return falls slightly by 11.1%, showing that an overly strict constraint will also affect the model. Moreover, we conduct visualization analysis in Figure 4 to demonstrate the significant effect on the distribution of prompts.

**(Q3) Behavioral Constraint.** Behavioral constraint is utilized in the second stage to enhance the model, which has a significant impact on the final effect. Similarly to Q2, we use the minimum coefficient of 0 and the maximum coefficient of 50 to show its impact. When the coefficient is 0, the average return will dramatically drop by 128%. Despite the fact that the loss decreases on offline datasets during the tuning process, the test results are still poor. This is the typical overfitting situation, indicating that the model suffers from severe data distribution shifts. When the coefficient is 50, the average return will also be reduced by 36.4%. It shows that an extremely severe behavioral constraint will also lead to inefficient policy boosting, resulting in slightly better performance than that of the model without prompt-tuning. Therefore, it is very important to find a suitable coefficient.

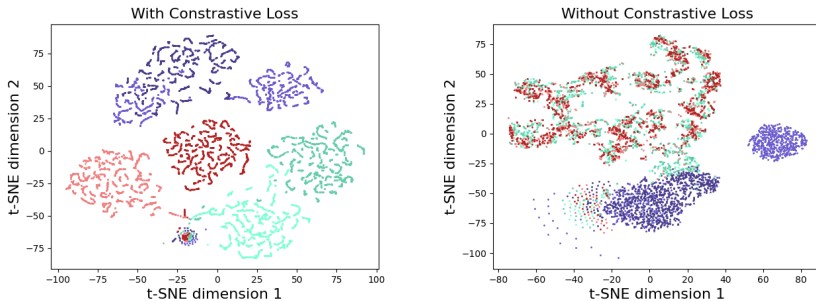

Figure 4: Visualization for Policy Prompts in halfcheetah task. We visualize prompts from three pairs of trajectories with contrastive loss and without contrastive loss. Each pair have similar behavior and reward sampled from offline reply dataset, and use similar colors.

**(Q4) Quality of task prompts.** Task content is constructed to accurately identify Meta-RL tasks. To demonstrate its effect, the experiments are divided into no context, medium context, and expert context groups based on content quality. In the first set of experiments, the CMT full model collects the context during online evaluation to support zero-shot adaption. The absence of task prompts significantly deteriorates the performance by 18% due to the inability to accurately identify tasks. The performance is improved when task contents from the offline datasets are employed. The results utilizing the medium and expert datasets increase by 4% and 7%, respectively. Both of them are better than the results in the first set of experiments. In fact, if offline task contents are used, the tasks will become few-shot tasks since the offline datasets are co-distributed with the tuning datasets. It is simpler and better than the zero-shot tasks using the online task contents.

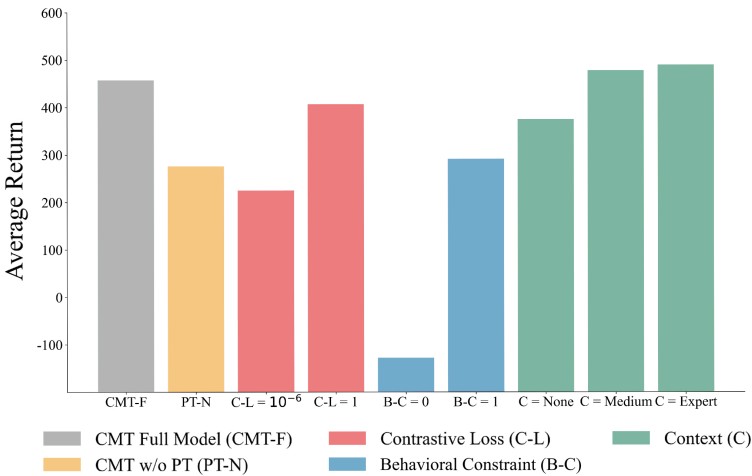

Figure 5: Results for ablation. We explore the impact of prompt-tuning, contrastive loss, behavioral constraint, and context on the model by whether use it or set extreme values. And each group has a different color. The first group is the full CMT model, which can be used as the benchmark.

# 6 CONCLUSIONS

In this paper, we present CMT, an offline RL algorithm based on prompt tuning, with the goal of training a large-scale model that can be utilized on various downstream tasks from the sequence modeling perspective. The prompt tuning is designed for offline RL to pre-train the model and guide the autoregressive model to generate trajectories with high rewards. Besides, a variety of experiments are conducted in three different RL settings, offline single-agent RL (D4RL), offline Meta-RL (MuJoCo), and offline MARL (SMAC), and the model's performance is evaluated with different baselines. The results show that CMT has strong performance, and generality. To our best knowledge, CMT is also the first sequence-modeling-based algorithm for offline meta-RL problems. General decision models like CMT enhance the efficiency of model training and lower the threshold for the applications of RL algorithms.

**Reproducibility Statement** To reproduce the results in this paper, our effort are listed as following:

- **Environment.** We give the detail information about our hardware platform in section 5. And the software version and dependencies are included in a requirement file.

- **Method.** In section4.1, we give all the loss function formula. The network architecture are shown in Figure 1 For the better understanding of the relationship of data and loss function, we illustrate the detail dataflow in appendix A.1. We also list all the hyper-parameter for the experiment in Appendix A.2.

- **Data.** Since the performance in offline RL highly depends on the dataset, we clearly report the dataset resource. In the offline RL setting, we adopt the popular benchmark (D4RL `https://github.com/Farama-Foundation/D4RL`). In the offline meta setting, we follow the FOCAL (`https://github.com/LanqingLi1993/FOCAL-ICLR`) to generate data.

- **Randomness.** To alleviate the randomness and noise, we run three seeds to reduce the bias. And we randomly sample the seed from a uniform distribution from $(0, 65536)$ without any selection.

- **Code.** Since we has not been completed code clean yet, we are glad to open source our code in more readability format in rebuttal.

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
