# OpenReview forum: "Contextual Transformer for Offline Reinforcement Learning"
_ICLR.cc/2023/Conference — Submitted to ICLR 2023_

### Official Review · Reviewer_SQ2X · 2022-10-25

**Confidence:** 5
**Correctness:** 3
**Technical Novelty And Significance:** 2
**Empirical Novelty And Significance:** 2
**Recommendation:** 5

**Clarity, Quality, Novelty And Reproducibility:**

Clarity: The paper is very easy to read.

Novelty: The individual "components" are known, and the way the paper tries to combine individual different components is very interesting.

Reproducibility: As "individual" components are well known or well "used" it should be easy to reproduce the proposed idea.

**Strength And Weaknesses:**

Strengths

- The paper is very well written, and motivated.
- The underlying idea is not "new" per se, but the way the idea is instantiated is interesting and very relevant. The paper also explores the idea in various different settings such as offline RL and offline MetaRL.
- The reviewer appreciate the ablations provided in the paper to dissect where the improvement is coming from.
- The reviewer appreciates the comparison with various different baselines for offline meta-RL tasks such as CBCQ, FOCAL, Batch-Pearl.

Weaknesses

- One way to improve the paper, is perhaps building on the offline MetaRL results. It's interesting to see that the performance of the proposed method is zero-shot as compared to other baselines which requires some context. Further evaluating the method on more complex tasks which requires adaptation as well as exploring how the adaptation varies by varying the nature of the data in the offline dataset would be very interesting.
- There are aspects of the work, which are already explored in the literature like learning a policy conditioned on the "past" experiences as done in Retrieval Augmented RL [1, 2]. These methods learn an encoding function using self-supervised learning, and then based on the context of the agent, figures out what information to retrieve to condition the current policy. The interesting difference is A_{g} in the current work is parameterized as  auto-regressive transformer which is trained via state, action and reward prediction, whereas in [1] it's parameterized as a neural network (non-LSTM) and trained with RL/behavior cloning (A_{e} is still parameterized as a bidirectional transformer.

Minor point:

- The reviewer is not sure if it's really important to mention the point ""CMT is the first algorithm to solve offline meta-RL from a sequence modeling perspective". Not sure what "solving" means here especially after only evaluating on Ant, Half-Cheetah, Walker.


[1] Retrieval-Augmented Reinforcement Learning, https://arxiv.org/abs/2202.08417 (ICML'22) ,\
[2] Large-Scale Retrieval for Reinforcement Learning, https://arxiv.org/abs/2206.05314 (NeurIPS'22)




**Summary Of The Paper:**

The paper proposes a method based on utilizing the "offline trajectories" as a latent variable (prompt), and then conditioning the behavior on the online agent on the latent variable. The paper also explores using a model based planner to achieve higher reward by adapting the latent variable which the paper mentions as "prompt tuning". The paper evaluates the proposed method across different settings: offline single-agent RL on the D4RL dataset, offline MetaRL on the MuJoCo benchmark.

**Summary Of The Review:**

The paper explores an interesting problem i.e., how to improve the policy given some offline data. The way the paper instantiates the proposed idea is interesting. It would be useful to evaluate the method on more complex tasks to further study the behavior of the proposed model.

---

### Official Review · Reviewer_7JkM · 2022-10-25

**Confidence:** 2
**Correctness:** 3
**Technical Novelty And Significance:** 2
**Empirical Novelty And Significance:** 2
**Recommendation:** 3

**Clarity, Quality, Novelty And Reproducibility:**

# Clarity

The paper is well-written and easy to follow

# Quality and Novelty

As far as I know, both the prompt design and meta-learning algorithms used in this paper are not new. The only novelty seems to be the combination of prompting and meta-learning for offline RL settings.

# Reproducibility

Although the authors provide the details like hyper-parameters in the Appendix, the authors did not provide their source code. They claim they will open source it in rebuttal.

**Strength And Weaknesses:**

# Strength

- Improving decision transformers with prompting is natural.
- The proposed prompt mechanism is well-designed.

# Weakness

- This paper failed to provide a complete literature survey. Especially, "Prompting decision transformer for few-shot policy generalization, ICML 2022” leverages the sequential modeling ability of the Transformer architecture and the prompt framework to achieve few-shot adaptation in offline RL, which is very similar to this paper’s approach. The authors of this submission should clearly explain what are the differences and properly cite this ICML paper.
- The performance of CMT doesn't seem to be impressive and robust. It would be better if the authors could conduct experiments to probe how performance varies according to different task prompts.
- Figure 5 is not clear enough for me. Fig. 5 should be segmented into 3 or 4 different tables or figures to separately investigate different hyper-parameters' influence. Only providing C-L=10^{-6}, C-L=1, B-C = 0, and B-C = 1 settings cannot completely show how these hyper-parameters influence the model.
- There is a performance gap between the proposed model and IQL. The performance should be further improved. Or the author should explain why the performance gap is acceptable at least.

**Summary Of The Paper:**

This paper investigates how to improve decision transformers with prompt and meta-learning. Evaluation on D4RL and MuJoCo benchmark tasks shows that the proposed method outperforms other baselines in some cases.

**Summary Of The Review:**

This paper investigates how to improve decision transformers with prompt and meta-learning. But the proposed method does not show significant and consistent improvements over other baselines. More importantly, prompting decision transformers has been introduced in an ICML paper before.

---

### Official Review · Reviewer_9AEp · 2022-10-27

**Confidence:** 4
**Correctness:** 3
**Technical Novelty And Significance:** 2
**Empirical Novelty And Significance:** 2
**Recommendation:** 3

**Clarity, Quality, Novelty And Reproducibility:**

The writing is hard to follow. Some statements are conflicted overall, e.g. in abstract, it is claim that this approach is training with supervised loss ('as such, we can pretrain a model ..... with supervised loss and .....'). While in Introduction, it claims that CMT is trained in a self-supervised manner (... Firstly, a model is pretrained on the .... through a self-supervised learning method). Based on my understanding, the training of CMT leverages two losses, where the autoregressive prediction is a supervised loss while the contrastive loss is a self-supervised loss. I would recommend the authors to keep the descriptions consistent in the whole paper. Otherwise, it is very confusing to the readers.  lack of implementation details and experiment settings. For example, as a pretraining work, I did not find the pretraining and finetuning datasets details, especially the coverage of tasks. Are you evaluating and/or finetuning on the same tasks or unseen tasks? Without such information, it is hard the validate the generalization ability of CMT.

The gains of CMT are marginal when comparing with other approaches.


The technical novelty is limited. From model architecture aspect, CMT is a combination of existing approaches, DT and bidirectional transformer. From algorithm aspect, it is an autoregressive loss (DT style) and contrastive loss. I can understand that this work is trying to propose a new pretraining regime, however, the usage is also not natural to me. Since the prompt  strategy can be directly utilized with only a causal transformer, in this case the DT. Then why an extra bidirectional transformer is needed? I did not find a clear answer in this paper. The rational under the contrastive loss is also unclear to me. For a specific task in a certain environment, the policy should be the same. However, based on the definition of positive and negative samples, it assumes that only the segments from the same trajectory share a policy while the other trajectories have different policies. I am confused about this setting, unless this contrastive loss are capturing other information or factors lie in trajectories, e.g. spatio-temporal information.

Minor:
There are no source codes are submitted.
Typo： Sec 4.2 paragraph 1, should be Fig. 2 not Fig (3).

**Strength And Weaknesses:**

Pros:
This work is well motivated which is exploring an interesting problem: how prompts can help sequence modeling for offline RL algorithms. To achieve this, a Transformer encoder-decoder architecture is presented, technically it is a combination of a bidirectional transformer and a causal transformer (DT). To pretrain such a model, two pretraining objectives are leveraged, i.e. autoregressive prediction and a contrastive loss as auxiliary loss during training. To evaluate the proposed approach, three different settings are been conducted, i.e. meta RL, offline RL and prompt tuning.

Cons:
1.the writing is hard to follow. Some statements are conflicted overall, e.g. in abstract, it is claim that this approach is training with supervised loss ('as such, we can pretrain a model ..... with supervised loss and .....'). While in Introduction, it claims that CMT is trained in a self-supervised manner (... Firstly, a model is pretrained on the .... through a self-supervised learning method). Based on my understanding, the training of CMT leverages two losses, where the autoregressive prediction is a supervised loss while the contrastive loss is a self-supervised loss. I would recommend the authors to keep the descriptions consistent in the whole paper. Otherwise, it is very confusing to the readers.
2. lack of implementation details and experiment settings. For example, as a pretraining work, I did not find the pretraining and finetuning datasets details, especially the coverage of tasks. Are you evaluating and/or finetuning on the same tasks or unseen tasks? Without such information, it is hard the validate the generalization ability of CMT.
3. The technical novelty is limited. From model architecture aspect, CMT is a combination of existing approaches, DT and bidirectional transformer. From algorithm aspect, it is an autoregressive loss (DT style) and contrastive loss. I can understand that this work is trying to propose a new pretraining regime, however, the usage is also not natural to me. Since the prompt  strategy can be directly utilized with only a causal transformer, in this case the DT. Then why an extra bidirectional transformer is needed? I did not find a clear answer in this paper. The rational under the contrastive loss is also unclear to me. For a specific task in a certain environment, the policy should be the same. However, based on the definition of positive and negative samples, it assumes that only the segments from the same trajectory share a policy while the other trajectories have different policies. I am confused about this setting, unless this contrastive loss are capturing other information or factors lie in trajectories, e.g. spatio-temporal information.
4. the gains of CMT are marginal when comparing with other approaches.

**Summary Of The Paper:**

This paper presents Contextual Meta Transformer (CMT) which is an offline RL algorithm based on prompt tuning. It targets on training a large-scale model that can be utilized on various downstream tasks from the sequence modeling perspective. The training of CMT consists of two stage, i.e. representation stage and improvement stage. The prompt tuning is designed for offline RL and guide the autoregressive model to generate trajectories with high rewards. Experiments are conducted in three different RL settings, offline single-agent RL (D4RL), offline Meta-RL(MuJoCo). and offilien MARL (SMAC).

**Summary Of The Review:**

This work is well motivated which is exploring an interesting problem: how prompts can help sequence modeling for offline RL algorithms. To achieve this, a Transformer encoder-decoder architecture is presented, technically it is a combination of a bidirectional transformer and a causal transformer (DT). To pretrain such a model, two pretraining objectives are leveraged, i.e. autoregressive prediction and a contrastive loss as auxiliary loss during training. To evaluate the proposed approach, three different settings are been conducted, i.e. meta RL, offline RL and prompt tuning.

However, there are some concerns:
1.the writing is hard to follow. Some statements are conflicted overall, e.g. in abstract, it is claim that this approach is training with supervised loss ('as such, we can pretrain a model ..... with supervised loss and .....'). While in Introduction, it claims that CMT is trained in a self-supervised manner (... Firstly, a model is pretrained on the .... through a self-supervised learning method). Based on my understanding, the training of CMT leverages two losses, where the autoregressive prediction is a supervised loss while the contrastive loss is a self-supervised loss. I would recommend the authors to keep the descriptions consistent in the whole paper. Otherwise, it is very confusing to the readers.
2. lack of implementation details and experiment settings. For example, as a pretraining work, I did not find the pretraining and finetuning datasets details, especially the coverage of tasks. Are you evaluating and/or finetuning on the same tasks or unseen tasks? Without such information, it is hard the validate the generalization ability of CMT.
3. The technical novelty is limited. From model architecture aspect, CMT is a combination of existing approaches, DT and bidirectional transformer. From algorithm aspect, it is an autoregressive loss (DT style) and contrastive loss. I can understand that this work is trying to propose a new pretraining regime, however, the usage is also not natural to me. Since the prompt  strategy can be directly utilized with only a causal transformer, in this case the DT. Then why an extra bidirectional transformer is needed? I did not find a clear answer in this paper. The rational under the contrastive loss is also unclear to me. For a specific task in a certain environment, the policy should be the same. However, based on the definition of positive and negative samples, it assumes that only the segments from the same trajectory share a policy while the other trajectories have different policies. I am confused about this setting, unless this contrastive loss are capturing other information or factors lie in trajectories, e.g. spatio-temporal information.
4. the gains of CMT are marginal when comparing with other approaches.

---

### Official Review · Reviewer_yV6V · 2022-10-31

**Confidence:** 4
**Correctness:** 4
**Technical Novelty And Significance:** 1
**Empirical Novelty And Significance:** 3
**Recommendation:** 6

**Clarity, Quality, Novelty And Reproducibility:**

Clarity: Very good.

Quality: Good. Would be stronger if the weaknesses above are addressed.

Novelty: Application to meta-offline-RL tasks is new. Methods are adopted directly from LLM community.

Reproducibility: Good.

**Strength And Weaknesses:**

Strength:
- The paper is well-written and easy to follow.
- Experiments are thorough, for offline, meta-offline, and ablation studies.
- The proposed method aligns well with existing literature

Weaknesses:
- Hyper-parameters have been tuned to improve performance, but the process has not been explained. It is unclear if there is over-fitting to test data.
- In LLMs, the size of the transformer model makes a large difference in performance. This has not been discussed or investigated.
- Prompt-tuning relies on beam search. This will only work well if the replay buffer has sufficient "good" data points. This is evident with poor performance on Medium-replay buffers. This should be analyzed and discussed.

**Summary Of The Paper:**

The paper proposes a transformer based method to solve offline meta reinforcement learning and offline reinforcement learning tasks. An auto-regressive transformer is learned on the existing trajectory, and prompt tuning is used to guide the prediction towards policy improvement. There is an additional contrastive loss term used to ensure better clustering properties.

**Summary Of The Review:**

Paper is good overall. It would be good to address the weaknesses listed in a revised version of the paper.

---

### Decision · Program_Chairs · 2023-01-20

**Decision:**

Reject

**Justification For Why Not Higher Score:**

There are clear issues of the paper to be accepted, including the consistency of the claims, the novelty of the technic, and the significance of the experiment results.

**Justification For Why Not Lower Score:**

N/A.

**Metareview: Summary, Strengths And Weaknesses:**

The paper proposes to use a transformer based method to solve meta offline RL and offline RL problems. Although all reviewers feel it is in principle reasonable, there are also concerns over the consistency of the claims, the novelty of the technic, and the significance of the experiment results.